# Inferred inflow forecast horizons guiding reservoir release decisions across the United States

Sean W.D. Turner[1], Wenwei Xu[1], Nathalie Voisin[1,2]

[1]Pacific Northwest National Laboratory, Seattle, WA, USA.
[2]University of Washington, Seattle, WA, USA.

*Correspondence to*: Sean W.D. Turner (sean.turner@pnnl.gov)

**Abstract.** Medium to long-range forecasts often guide reservoir release decisions to support water management objectives, including mitigating flood and drought risks. While there is a burgeoning field of science targeted at improving forecast products and associated decision support models, data describing how and when forecasts are applied in practice remains 10  undeveloped. This lack of knowledge may prevent hydrological modelers from developing accurate reservoir release schemes for large-scale, distributed hydrology models that are increasingly used to assess the vulnerabilities of large regions to hydrological stress. We address this issue by estimating seasonally-varying, regulated inflow forecast horizons used in the operations of more than 300 dams throughout the Conterminous United States. For each dam, we take actual forward observed inflows (perfect foresight) as a proxy for forecasted flows available to the operator, and then identify for each week of the year 15  the forward horizon that best explains the release decisions taken. Resulting "horizon curves" specify for each dam the inferred inflow forecast horizon as a function of the week of the water year. These curves are analyzed for strength of evidence for contribution of medium to long-range forecasts in decision making. We use random forest classification to estimate that approximately 80% of large dams and reservoirs in the US (1553±50 out of 1927 dams with at least 10 MCM storage capacity) adopt medium to long-range inflow forecasts to inform release decisions during at least part of the water year. Long-range 20  forecast horizons (more than six weeks ahead) are detected in the operations of reservoirs located in high elevation regions of the Western US, where snowpack information likely guides the release. A simulation exercise conducted on four key Western US reservoirs indicates that forecast-informed models of reservoir operations may outperform models that neglect the horizon curve—including during flood and drought conditions.

## 1 Introduction

Dams regulate nearly all rivers in the United States. They generate more than half of US renewable electrical power, protect thousands of communities against damaging floods, and supply copious water for the nation's irrigated agriculture and urban water systems (US Army Corps of Engineers, 2015; Bureau of Reclamation, 2016). To provide these essential services, dams must be operated efficiently for uncertain hydrological conditions days and weeks ahead. Water managers thus rely increasingly on reservoir inflow forecasts to guide water release decisions (Gong et al., 2010; Brown et al., 2015; Boucher and

Ramos, 2018)—and will continue to do so as the range, resolution, and quality of hydrological forecast products continue to advance (e.g., Wang and Robertson, 2011; Yuan et al., 2015; Bennett et al., 2016). Inflow forecasts are valuable because they help operators manage difficult trade-offs. For example, the threat of drought is best addressed with maximum stored water, while the threat of flooding requires spare storage capacity for capturing water. Knowledge as to the likelihood of either hazard is thus indispensable when deciding how much water to hold in storage. This is why, for instance, the depth of upstream winter

snowpack in high Western US headwaters, which provides a strong indication of the volume of water likely to enter a reservoir in the spring, guides operators on how much water to hold in storage early in the year (Garen et al., 1992).

While we know that inflow forecasts are useful, our understanding of their precise contribution to water release decision making across a large number of dams is limited. Lack of detailed reporting of operational rules and guidelines means the science community remains largely uniformed on a number of key details, such as typical forecast lead times adopted, and

times of year when forecasting is deemed most important. To our knowledge, these data have yet to be collected through any qualitative or quantitative research study conducted at national scale. Lacking accurate operational data and associated decision-making schemes, large-scale, distributed hydrology models (e.g., Van Vliet et al. 2016, Wada et al. 2016; Voisin et al. 2013b; Voisin et al., 2017; Vernon et al., 2019; see Nazemi and Wheater, 2015 for a review) are liable to misrepresent the influence of human water management on river flows (Yassin et al., 2019), including during extreme flood and drought

conditions. The applications of these models—increasingly, large-scale multisectoral planning studies aiming to predict stresses on water, energy, and food systems—may in-turn suffer mischaracterization of human systems' exposure to hydrological risks.

This paper asks whether the use of forecasts in real-world operations can be inferred (i.e., back-calculated) from historical records of reservoir storage, inflow, and release. We suggest that the contribution of forecasts to decision making at a given

dam can be described quantitatively through construction of seasonally-varying estimates of regulated inflow forecast horizons adopted by the operator (herein termed the dam's "horizon curve"—a novel concept introduced in this paper). To test this hypothesis, we attempt to infer the horizon curves and associated water release policies for a sample of 316 dams and reservoirs in Conterminous United States. Since horizon curves are derived empirically for each dam using observed (i.e., regulated, non-natural) inflows and releases, the estimated horizons are attributed to a water forecast that could be derived from any source

of information, including meteorological, climatological, and hydrological predictions, as well as knowledge of planned water management, such as scheduled releases from a large dam upstream. The approach is therefore agnostic to the possible sources of information that an operator may deploy to predict future inflow. We explore how the inferred horizon curves vary across dams, and then interpret some of the dominant as well as unexpected operator behaviors by focusing on particular cases. By labeling each dam's horizon curve according to whether or not it provides compelling evidence for medium-range inflow

forecast use in operations, we identify (using random forest classification) the dam and reservoir characteristics that are conducive forecast-informed operations. Finally, simulations at four key dams are used to test whether the horizon curve could lead to improved representation of water management in large scale hydrological models.

## 2 Method

### 2.1 Justification for the concept of a horizon curve

Several factors determine whether and how foresight informs water release decisions, and these factors vary widely across dams. For example, the value of an inflow forecast may depend on the characteristics of the reservoir. There are diminishing returns in low-memory reservoirs (low storage capacity relative to inflow) and for certain operating purposes (Georgakakos and Graham, 2008; Graham and Georgakakos 2010; Zhao et al., 2012; Anghileri et al., 2016; Turner et al., 2017). If the reservoir characteristics are suitable, the operator's decision to adopt a forecast-informed release policy will then depend on

perceived forecast reliability and how that reliability varies throughout the year (Rayner, 2005; Whateley et al., 2014). Forecast reliability, in turn, depends on the available predictive information. An operator might rely on upstream water storage (e.g., soil moisture, snowpack, lake levels) (Shukla and Lettenmaier, 2011), hydrological regime state (Turner and Galelli, 2016), climate indices and teleconnections (Yang et al., 2017; Libisch-Lehner et al., 2019), weather forecasts (Georgakakos et al., 2005; Shukla et al., 2012; Nayak et al., 2018), current river flow rates (Hejazi et al., 2008), knowledge of planned water releases

from upstream dams, and perhaps some or all of these in combination (Denaro et al., 2017). This enormous scope for variability in forecast quality and application across dams means there is no obvious way to identify the actual operationalized forecast, or indeed the model used to assimilate it into decision making, for a given system without insight into individual agencies' models and data preferences. Large-scale hydrology models may encompass several hundred dams, so acquiring this insight through qualitative survey would be a major challenge. We therefore propose a practical, empirical approach to inferring—or

back-calculating—seasonally-varying forecast horizons adopted in dam and reservoir operations.

### 2.2 Derivation of horizon curves

To characterize the contribution of forecasts to release decisions across a large sample of dams, we adopt a simple, regression-based method that can be applied to any reservoir for which observational daily time series of storage volumes and at least one of inflow or release are available. The approach returns for any dam a signature of the inferred regulated inflow forecast

horizon over the water year (the horizon curve) as well as an associated inferred operating policy describing how future inflow out to those horizons informs the release and depending on time of year. To achieve this, we must first assume that the future observed inflows (perfect forecast) may act as a proxy for the actual forecast available to the operator at the time of deciding how much water to release (the limitations of this necessary assumption are covered in the Discussion section). If this holds, and if regulated inflow forecasts contribute substantially to the release decision amidst other rules and constraints, then the

future inflow would better indicate the observed release decision than the current inflow. In other words, the operational forecast horizon is assumed to be the one that best explains the release decision taken. No prior assumption of forecast use is needed, because the method identifies for each dam whether a forward inflow horizon substantially informs the release decision.

For a given dam or reservoir, the procedure is executed as follows (as illustrated in Figure 1). First, daily time series of inflow and release rates are aggregated to weekly volumes (in million cubic meters, MCM) by water week, with week 1 starting 1st October—the start of the hydrological year. The weekly timestep allows us to reasonably back-calculate inflow or release (if either is missing) from change in storage, using conservation of mass and assuming negligible evaporation and other losses (such estimates are not reliable at the daily scale because storage and flow variables tend to be reported as daily averages). This gives us a much larger sample of dams to work with. Then, for each water week, the interannual values of starting storage, release, and inflow are used to fit release-availability functions (i.e., relationships that specify the water release decision as a function of water available) for multiple candidate future inflow horizons, where in each case the availability, $a$, is computed as the starting storage plus the cumulative future inflow out to horizon $h$ weeks (see example in Figure 1). The inferred release-availability function fitted to these data is a piecewise linear model with a single breakpoint. This model provides appropriate flexibility to account for the typical behavior of operations, wherein an excess of incoming water may be addressed with a comparable increase in the release (leading to a relatively sharp positive slope on the right-hand side of the function), while a lack of water must be satisfied by a reduction in the release, which cannot be negative and which is often bound by a requirement to provide environmental flows (leading to a flat or low-gradient positive slope on the left-hand side of the function). These piecewise functions are fitted to observed release and availability data for all possible horizons in increments of one week (with x-axis availability, $a$, recomputed for each horizon). Functions may be fitted for a given set of breakpoint coordinates by performing linear regression on either side of the breakpoint; the breakpoint coordinates themselves can be optimized for minimum sum of squares in the model residuals. Once functions are fitted for all candidate horizons, the horizon offering the closest fit to observed decisions is selected as the operational horizon for that water week of the year (subject to some conservative adjustments described below).

A desirable feature of this model is its compatibility with archetypal rule curves often implemented in practice. To demonstrate this compatibility, let us first assume that forecasts are not used at all and that the operator uses seasonally-varying storage targets to decide on how much water to release. Our scheme allows us to mimic this type of operation very effectively. The two-stage piecewise model provides the flexibility to represent two situations: (a) the reservoir is above the guide curve, in which case one would expect the operator to respond by releasing water to draw the reservoir back down (with higher release for higher storage levels—allowed for by the right-hand slope of the piecewise function); and (b) the reservoir is below the guide curve, in which case the operator would wish to cut back the release significantly to allow the reservoir to refill. The breakpoint of a given week's piecewise function on the horizontal axis (which, absent the forecast, is simply water in storage) essentially represents the traditional reservoir guide-curve level, and the slopes either side of the function specify how the operator behaves in either situation. Our scheme is of course also designed to allow us to integrate forecast information in the rule-curve system. The difference is that the rule is no longer a function of current storage, but becomes a function of current storage plus the inflow out to various candidate horizons.

A conceptually similar piecewise model is presented in Yassin et al. (2019) with the fundamental difference that the process omits forecasts and is therefore based on climatology and current storage and inflow conditions. Importantly, because our algorithm is also computed for individual weeks, it ensures that the effects of operational decisions driven by long term average water availability conditions are intercepted and removed when estimating the horizon. For example, a simple operating rule designed to accommodate typical high flows delivered in springtime would not lead to detection of a forecast horizon in our procedure. The reason is that such actions are a function of time of year rather than forecasted flow. High releases to create a flood pool in anticipation of typical springtime flows can be implemented in reservoir rules without foresight, for example by releasing high volumes of water each March. Foresight would only be detected if there were clear evidence of the flood buffer being adjusted based on a forecast of the inflow and its deviation from normal conditions for that time of year. This behavior has been confirmed through a simple simulation exercise to check for the unwarranted detection of horizons beyond current-period in a synthetic reservoir with a seasonally-varying operating rules that are not explicitly forecast-informed.

## 2.3 Experimental setup

For this study we compile daily observed storage, inflow and release time series for more than 900 dams and reservoirs in Conterminous United States (sources are US Army Corps of Engineers, US Bureau of Reclamation, US Geological Survey, California Data Exchange, and Texas Water Development Board; see acknowledgements). In cases where only storage data are available, releases are obtained from USGS streamflow gauges immediately downstream of the dam. After addressing minor gaps (ten continuous days or less), we remove incomplete, short (less than ten years' continuous data) and duplicate records, leaving a set of 316 dams with sufficient data for creating horizon curves. These dams represent a range of operating purposes and reservoir storage sizes and are well distributed across the Conterminous United States west of the Mississippi River (Figure 2). We then create a horizon curve for each dam following the steps outlined above and in Figure 1. Piecewise functions are fitted to each water week (1, 2, … 52) and future inflow horizon (1, 2, … 30 weeks) combination for each dam (total of $52 \times 30 = 1560$ functions fitted for each dam) by identifying the function breakpoint coordinates that minimize root mean squared error, achieved using a numerical optimization algorithm designed for derivative-free, non-linear problems (Powell, 2009) and found to perform efficiently in our testing. To avoid overfit to unrealistic operating policies, the piecewise functions are constrained such that both slopes are non-negative and that the right-hand slope exceeds the left-hand slope. We wish to avoid inferring forecast contribution in cases where the evidence is marginal relative to lower horizons or no-forecast cases. We therefore infer forecast contribution only when the policy for forecast horizon $h$ results in a substantially better fitting policy ($> 0.1$ increase in $R_2$) relative to forecast horizon $h - 1$. In other words, when the strongest policy fits are similar across a range of horizons (say, an increase in $R_2$ of less than 0.1 between horizons of 6 and 7 weeks ahead), the lowest of these horizons (6 weeks ahead) is assumed to drive the release decision. Given the imperfections of the process, a degree of noise is to be expected in the derived horizon curve for any dam. This is addressed by de-spiking each horizon curve and then

smoothing using a locally-weighted smoothing spline (Cleveland, 1981). All of these calculations and assumptions are made freely available through an open code repository ([www.github.com/IMMM-SFA/horizon](www.github.com/IMMM-SFA/horizon)).

### 2.4 Classification of horizon curves

With the horizon curves derived, we use Random Forest classification (Ho, 1995) to identify features of dams associated with detection of significant medium to long-range inflow forecast contribution in the horizon curve. The classification not only helps us interpret results but also allows for exploring the possibility of extrapolation of horizon curves to dams with no observed data. In this analysis the target response variable is a simple Boolean (true/false) of whether there is evidence for significant forecast contribution or not. This first requires a definition of what constitutes significant forecast detection in the horizon curve. One can anticipate that many horizon curves could contain only very weak evidence for foresight in operations (e.g., a horizon curve in which periods of apparent forecast-use are sporadic and with short lead times). Ideally, these should be labelled as non-significant horizon curves. Unfortunately, separating these low-evidence cases from the others is a rather arbitrary exercise. We label a horizon curve as significant (meaning containing sufficient evidence for medium-to-long range forecast horizons) if it contains an unbroken, three-week period with operating horizons of at least two weeks ahead, and with the coefficient of determination of the release-availability functions associated with those horizons exceeding 0.5. Relaxing these thresholds would of course result in more dams being categorized as significant (and vice versa), so we take the additional step of performing sensitivity to changes in these thresholds (with relevant results included in Appendix A).

The candidate explanatory features in this classification analysis include dam and reservoir specifications, operating purposes, and various statistics describing the inflow and storage time series (variability, autocorrelation, etc., at various time scales)— a total of 26 features (listed in Table A1). It's essential that a Random Forest classification scheme is set up to avoid the possibility of overfitting. Here we set the number of trees to 1000 and limit the number of decision layers to a maximum of three. A bootstrap is used to repeat Random Forest generation 200 times with different training-test splits. In each case, a different random sample of 70% of dams constitute the training set while the remaining 30% are used as unseen testing data for validation and evaluation. Feature importance is then determined using the commonly used Gini impurity of the tree. Gini impurity, like entropy, describes the likelihood of an incorrect classification. The feature importance score is calculated as the percentage decrease in Gini impurity with that feature included, averaged across the forest (see Friedman et al., 2001).

### 2.5 Practical application of the horizon curve in a reservoir simulation

The intended application of the horizon curves is to enhance reservoir release schemes of large-scale, distributed hydrological models incorporating water management specifically in anticipation of flood and drought events. The derivation of horizon curves involves determining the horizon that leads to the best-fit operating policy (release as a function of available water) at weekly intervals, leading to a relatively high-resolution dataset that could be deployed in these models. A regional-scale hydrological simulation lies beyond the scope of the current study and is being conducted in ongoing research. Nonetheless,

we can explore the potential improvements that a forecast-driven model might proffer by conducting offline, single reservoir simulations forced with observed inflows. For each of four significant, large storage dams in the Western United States, we perform the horizon curve derivation procedure as outlined above. The chosen dams are Grand Coulee (Columbia River), Glen Canyon (Lake Powell, Colorado River), Shasta (Sacramento River, California), and Dworshak (Snake River), which represent a set of diverse storage capacity, flow seasonality and level of regulation in inflows. For each water week and selected horizon there is an associated piecewise function that specifies a release decision as a function of water availability (storage volume plus cumulative future inflow out to the duration of the horizon). Simulation is performed using observed inflow and storage levels. This means errors in the release are not allowed to accumulate through storage, providing the cleanest test of overall decision accuracy across all data points. Further testing that allows storage error accumulation and includes the effects of inflow bias lies outside the scope of this work and is being conducted in ongoing research. To compare results against a release policy that neglects forecasts, we use the same format of piecewise models but instead train them with a uniform horizon of one week ahead for all weeks of the year. This means the benchmark operating policy is a function of current storage plus current inflow (with the model varying by week).

## 3 Results

### 3.1 Horizon curves for 316 dams

We group resulting horizon curves according to the timing of peak horizon (i.e., the week of the year when the maximum forecast horizon is used) within the water year and then order within each group by magnitude of peak horizon. Resulting horizon curves are displayed in Figure 3. Of the 316 horizon curves derived, use of foresight is detected in 283 cases (i.e., 283 cases in which at least one week of the year contains a detectable horizon of at least one week ahead of the current week). This equates to 90% of dams studied. The remaining 33 dams have completely flat horizon curves—suggesting that the releases from these dams are guided at all weeks of the year using information on currently available water alone. Perhaps surprisingly, the timing of the peak horizon varies widely across dams. Horizon peaks occurring toward the end of the water year tend to be short-lived, lasting three or four weeks. In contrast, horizons detected in earlier in the year (from weeks 9 through 25, or mid-December through early April) are often drawn out, lasting a number of months.

While no two horizon curves are identical, some consistent and intelligible patterns emerge. Flat horizon curves with consistent horizons of one week (which should be interpreted as the current period inflow—or no forecast use) and consistently strong policy fits ($R_2 > 0.9$) are found for run-of-river hydropower facilities, such as Ice Harbor Lock and Dam, on the Columbia River, Washington (Figure 4a). These dams have very low storage relative to inflow (typically a day's flow or less), meaning forecasts of more than a few days ahead are superfluous. At a weekly resolution, inflow is close to outflow, so we observe a near-perfect relationship between release and current water availability, and progressively weaker relationships as the horizon

is extended. Though unsurprising, this result is satisfying because it demonstrates that cases where forecasts certainly do not influence the decision are easily identified as such by the derivation procedure.

Evidence for week-ahead horizons begins to emerge as we move to reservoirs with slightly longer memory (Figure 4b). Orwell Dam, Minnesota, for example, impounds a small, upstream reservoir (~25 MCM) used for flood control and municipal water supply. Storage capacity is about five percent of annual inflow. Here we infer week-ahead forecast use during a few periods of the latter half of the water year. The region is prone to summer thunderstorms, so perhaps severe weather warnings during these weeks have, on occasion, prompted operators to lower reservoir levels to increase the flood buffering volume.

In cases where long-range forecasting is inferred (defined here as four consecutive water weeks with a horizon of six weeks or more), horizon curves tend to be n-shaped: low during the beginning and end of the water year, with a significant rise emerging in winter or early spring and then fading off by early summertime. These cases are indicative of snowpack driven forecasting. Operations at Glen Canyon Dam (Lake Powell) exemplify this behavior (Figure 4c). Here we observe inferred forecast horizons increasing rapidly by the start of the calendar year—neatly coinciding with the first issue of April-July
streamflow forecasts provided by the Colorado River Basin Forecast Center—and then slowly declining in horizon as the snowmelt season approaches. Similar examples include Jackson Lake, Wyoming (Figure 4d), and Bridgeport Reservoir, California (Figure 4e), for which the inferred horizon rises at the onset of the snowmelt season (early April) for the Rockies and the High Sierra, respectively. Perhaps in these cases early-year forecasts are too unreliable to inform releases. Or perhaps early-year forecasts do inform releases, with the policy undetected here due to the uncertainties or conservative assumptions
embedded in the derivation procedure (e.g., use of actual observed future flow instead of the actual forecast available to the operator).

Some horizon curves require more in-depth interpretation. A few dams follow the same snowmelt-driven forecast behavior described above, but also appear to use significant foresight during fall (i.e., at the start and end of the water year). This may indicate use of seasonal water outlooks informed by ENSO, which improves the skill of winter precipitation forecasts in the
region (Yang et al., 2018). Canyon Ferry, Montana (Figure 4f), and Millerton Lake, California (Figure 4g), are two such cases, although we must be careful not to conflate climate forecasts with other possible sources of foresight. Millerton Lake lies below a cascade of dams on the San Joaquin River; coordinated operations, rather than hydrological forecasting, may provide the foresight to guide releases. Indeed, it appears that in other cases the guiding information comes not from any hydrological or meteorological forecasts, but from simple knowledge of planned upstream water management decisions. Agate Dam and
Reservoir (Figure 4h) depends almost exclusively on diverted water from upstream storage via a canal system. Close inspection of release decisions reveals very clear correlations between release and future inflow at specific points in the water year. The January release is typically zero, with the two exceptions: 2002 and 2017. For both years the currently available water at the time of those releases is normal, but the cumulative future (diverted) inflow is well above average, suggesting that releases from this dam are closely coordinated with planned upstream diversions. Generally, we may assume that if a dam depends

almost entirely on the water management decisions from upstream reservoirs, and if those decisions can be planned weeks ahead in advance, then the inflows can be known with a high degree of accuracy and could be used to guide decisions. Knowledge of upstream water management decisions (either dam releases or perhaps planned abstractions for irrigation or other purposes) rather than hydrological or meteorological forecasting may explain much of the operational foresight detected in summer months at several dams (week 40 onwards in Figure 3).

Correlation between current release and future inflow need not always imply that the release is driven by knowledge of future inflow. It could be that the future inflow is driven by the release. Suppose for instance that a dam is called upon to release significant volumes of water over an extended period of time to address water quality concerns, resulting in a significant drawdown below the reservoir guide curve. This release event could trigger an upstream operational response to refill the downstream reservoir, perhaps over a period of several weeks. Complex coordinated operations of this sort are bound to create a myriad of uninterpretable wrinkles in the horizon curves derived. These complexities highlight the enormity of the challenge faced large-scale hydrological modelers trying to represent human water management actions without information on the actual operating schemes deployed in practice.

### 3.2 Features of dams with significant horizon curves

We applied the significance test described in section 2.2 on the horizon curves. Of the 316 dams studied, 256 (82%) are classified as having a significant horizon curve after applying these criteria. Relaxing these thresholds would of course result more dams being categorized as significant (and vice versa), so we take the additional step of performing sensitivity to tightening and relaxing of thresholds (results presented in Appendix A). After applying the classification scheme described in section 2.3, dam and reservoir features that best determine whether the horizon is significant are: the storage ratio of the dam (storage over mean annual inflow), the annual inflow volume, the average timing (within the water year) of minimum reservoir storage, dam elevation above sea level, and variability of storage and inflow time series at interannual and seasonal resolution (Figure 5). The storage ratio determines the memory of the system; forecasts do not contribute to release decisions for reservoirs with low storage ratio, as reported above with respect to run-of-river hydropower dams. As such, the storage ratio— and related features such as mean annual inflow and storage capacity—are among the most important variables in determining horizon curve significance (Figure 6a). Neither the dam's primary purpose (water supply, hydropower, irrigation, flood control, etc.) nor the source of data (US Army Corps, US Bureau of Reclamation, etc.) provide predictive capability in the random forest classification scheme.

Features describing water week with minimum storage, within-year variability of storage levels, and dam elevation may all be significant because they indicate the likelihood of a snowmelt driven regime. Spring snowmelt reservoir refill patterns are typical of high elevation dams (Giuliani and Herman, 2018). Snowpack volumes are the most reliable source of long-range streamflow forecast skill in snowmelt-dominated Western United States (Day 1985, Pagano et al., 2014), so one should expect that features like elevation become more important in determining whether long range forecasts contribute. This indeed appears

to be the case. If we group horizon curves into separate categories based on the longest forecast horizon observed, we find that long-range forecasts (six to eleven weeks ahead) and seasonal forecasts (twelve weeks ahead or more) typically contribute to the release decisions of high elevation dams and reservoirs (>1000 meters above sea level) (Figure 6b). Long range and seasonal horizons are found in approximately 35 % of dams with elevation below 500m above sea level, compared with 46% of dams in the 500 – 1000m category, and more than half of dams in the 1000 – 1500m and > 1500m categories. Corroborating this finding, in the months leading up to the snowmelt season (weeks 9 through 25, Figure 3) we observe prolonged inferred horizons that reflect the long period of snowpack accumulation during which long-range foresight is available.

The random forest classification scheme can be used to infer whether or not dams and reservoirs outside of the study sample are likely to apply medium to long-range forecast horizons. To extrapolate our results across all large dams (greater than 10 MCM storage capacity) in the Conterminous United States (1927 large storage dams in total), we re-train the Random Forests classification model using features that are available for all dams represented in the Global Reservoir and Dams (GRanD) database (Lehner et al., 2011). To this we add two additional features describing the number and accumulated storage of upstream dams (created from watershed mapping). Data describing the variability and autocorrelation of the inflow and storage time series are unrepresented in the GRanD, so must be excluded from the classification model. This turns out to be unproblematic; a Random Forest trained with only the storage ratio, elevation above sea level, mean annual inflow, storage capacity, and number of upstream dams is sufficiently accurate in validation, with strong scores of 0.91, 0.89, and 0.94 achieved for the common accuracy metrics of F1 score, precision and recall, respectively (see Appendix for mathematical definitions of these scores). The fact that these scores are achieved without the additional features suggests that these features may be redundant with others represented in the pared-down feature set. We use this pared down model to extrapolate our results for all dams in CONUS. The classification model estimates that $1553 \pm 50$ (90% confidence interval), or 82%, of large dams (storage capacity > 10 MCM) are characteristic of dams with significant horizon curves. Inferred horizons are prevalent across large dams. Approximately 81% of CONUS dams with storage capacity greater than 100 MCM are estimated to have releases influenced by inflow forecasts; for dams with storage capacity greater than 1000 MCM (139 dams), the estimate is about 90%. Regions where inflow forecast-contribution is prevalent include mountainous regions of CONUS, such as the along the spine of the Rocky Mountains, the Sierra Nevada of California, Cascades of the Pacific Northwest, and the Appalachians to the east (Figure 7).

### 3.3 Improvements in reservoir simulations using the horizon curve

Figure 8 displays policy simulation results for Grand Coulee, Glen Canyon, Shasta and Dworshak dams. The simulations are driven by actual observed inflow in each case. For each dam, results are shown for two simulations: simulated optimized piecewise policies assuming release to be informed only by current water availability, and simulated optimized piecewise policies using future flow as defined by the inferred horizon curve. These results demonstrate significant improvements in release decisions (relative to observation as measured by root-mean-squared error, RMSE) for the daily simulation, annual

daily maxima, and annual average 90-day minima  time series of releases, as well as for the transformed RMSE (TRMSE), wherein the simulated and observed releases are first transformed so that the result is weighted by performance during periods of low release (Box-Cox transform with exponent of 0.3, as adopted in van Werkhoven et al., 2009) (Table 1). This, and the maxima and minima assessments, are added to indicate performance improvements during flood and drought conditions. While some of these improvements are marginal (5 – 10% reduction in RMSE), one could hypothesize that there would be substantial differences in the representation of regional water management if such improvements were repeated across a large sample of dozens or perhaps hundreds of dams. Moreover, a marginal difference in a reservoir's capability to release or store water during an extreme event could imply a substantial difference in the downstream impact.

## 4 Discussion

The water management modules of large-scale hydrology models have to-date relied on relatively simple heuristics to simulate releases, such as monthly storage and release targets based on average climate (Hanasaki et al., 2006; Döll et al. 2009, Biemans et al., 2011; Solander et al., 2016; Voisin et al., 2013, 2017) or year-ahead, perfect foresight (Haddeland et al. 2006). Important nuances, such as the appropriate environmental release, are typically applied uniformly across all dams. The parameters of the 52 (weekly) piecewise release-availability functions (including detail of the forecast horizon) could inform a far more detailed and representative set of operating schemes with forward looking operations. This could be crucially important in many regions where inflow forecasts greatly enhance the reservoir's capability for flood and drought alleviation. Given the prevalence of forecast application, as suggested by this study, improved dam and reservoir models that represent intelligent operator response to anticipated reservoir inflows over seasonally-varying horizons within the myriad of other operational constraints should contribute to a better understanding of hydrological stressors on energy and food security that are increasingly linked to large-scale hydrological models (e.g., Van Vliet al. al. 2016, Wada et al. 2016; Voisin et al. 2013b; Hejazi et al. 2015, Voisin et al., 2018).

This approach to deriving a horizon curve is clearly not without limitations. Streamflow forecasts used in practice are often highly uncertain, so strong correlations between release and actual future observed inflow may be elusive, particularly for long-range forecast horizons. In theory, this issue could be addressed by using the actual forecasts available to operators to inform the availability axis of the candidate release-availability functions. In reality, these data are difficult to acquire—particularly for large-scale studies with many dams and reservoirs. Often the forecasts adopted will be probabilistic in the form of a forecast ensemble, rather than a deterministic forecast. In the present work, the actual observed future inflow is an imperfect yet practical alternative, and the results obtained suggest that it can be effective in many cases. Another challenge is selecting the correct study period. Ideally, a multi-decadal time series would be used to capture inter-annual variability in release and water available for all periods of the water year. The flipside is that the operating policy may have changed at some point in the last few years—it may be that new forecast products were introduced only the latter years of record, for instance.

In such cases of non-stationarity in the policy it would be prudent to use only those latter, forecast-informed years of operation so as to avoid averaging away the forecast-use signal. Lacking prior knowledge of how or when forecasts may have been introduced, the practical approach is to discard operations prior to some cut-off year (in the present work we use 1995, but also test the robustness of this decision using cut-off years of 2000 and 2005). Non-stationarity in the inflows is not a limitation here; as long as the operating policy is consistent through time, then a wide range of possible inflow conditions would be

desirable for determining the nature of that policy. A related problem is that the resulting models are not conducive to the type of rigorous validation exercise that has become standard in hydrological study. Apart from the problem that we are uninformed as to whether the policy of a given reservoir may have changed radically during the period of record, there are simply too few data points to support robust validation (~20 data points for 20 years of data, in a good case, which will contain perhaps only one or two flood or drought years to guide either side of the release-availability function). In the absence of long records of

consistently-applied policy, it's vital to protect against over-fitting. We achieve this by constraining each piecewise function to an expected, archetypal form (see 2.3 Experimental setup), although the corollary is that the resulting functions may in some cases be over-constrained. They may lack the required flexibility to represent more complex operating rules applied in practice. Despite these limitations, we find that the approach arrives at convincing evidence for regulated inflow forecast contribution as well as a range of other interesting operator behaviors. While the associated release policies are likely to be highly imperfect

models of actual operations, they potentially offer a significant advance on general, theory-driven rules currently adopted in state-of-the-art large-scale, distributed hydrological models (see Yassin et al., 2019, for a state-of-the-art review of existing approaches).

## 5 Conclusions

The use of foresight in reservoir release decisions can be interpreted without reported operating rules for individual dams and

365 reservoirs. All that is needed is operational data—time series of storage and flows into and out of reservoirs—and an appropriate release-availability function that can be fitted to these data to test a range of candidate operational horizons. Our analysis is the first to use this idea to estimate the contribution of regulated inflow forecasts to reservoir releases across a large number of dams and reservoirs. The results provide a first national scale estimate of the existing contribution of monthly to seasonal flow forecast to release decisions. The general approach of horizon curve derivation is inhibited by a number of non-

370 trivial challenges. These include identifying the appropriate operational period from which to build the curve, reconciling the differences between the forecast used by the operator and actual inflow over the horizon, and selecting appropriate thresholds for the indicating evidence for foresight contribution, such as the goodness-of-fit of the release-availability function. The single-breakpoint piecewise function adopted in this study is simple, but intelligible and, most importantly, effective for the purpose of identifying release policies driven by foresight of future inflow. And although the exactness of the horizon curves

is undoubtedly impaired by the limitations noted above, our analysis supports some interesting conclusions.

First, we find that the use of operational foresight—determining what to do in the present with some foresight of what will happen in the coming weeks or months—is prevalent in US dam and reservoir management. We detect a significant contribution of regulated inflow forecasts of at least one week ahead in more than 80% of dams in our sample of 316 dams. A similar proportion is estimated when we extrapolate to a much larger sample of CONUS dams with capacity greater than 10 MCM. Second, our classification exercise highlights the potential to extrapolate horizon curves to data sparse reservoirs. Large dams and dams at high elevations appear more likely to adopt longer range forecasting, but aside from the general and obvious rule that run-of-river facilities cannot benefit from forecasts, it appears that dams of all sizes, purposes, and locations rely on some degree of medium-range foresight to guide operations. Detected foresight appears to derive from a wide variety of sources, including climate and weather forecasting, but also from coordinated operations between dams. Some particular patterns, such as snowmelt forecasting, are intelligible from the horizon curve shapes and the dam features (e.g., high elevation). The importance of forecasting to release decision making may be studied in future research to understand the role of rule curves, forecast accuracy, reluctance to adopt forecast into operations (see Rayner 2005), and other factor that may limit the value of forecast to release decision making. Whether a more detailed and accurate approach to identifying the source of information leading to forecast can be derived from operational data alone is also a challenge for future research. Classification models, such as Random Forests, may be useful for extrapolating not only the presence of a significant horizon curve, but also parameters of policy functions for reservoirs lacking the operational data to build a policy directly.

Our approach, as configured in this work, assumes that operators use release-availability functions based on cumulative forecasted inflow. In reality the forecasts may be assimilated in a different way. For example, many reservoir operators follow rule curves and release water according to a step-wise function, or they may deploy a threshold-based forecast across a range of horizons. While our approach is simple and intuitive, the integration of forecasts into decision making is a complex process and all subtleties might not be captured. Our study may motivate further work at a national scale into understanding how forecasts are integrated into decision making by dam operators. Application of horizon curves and their associated release-availability functions in regional-scale hydrological modeling is being tested in ongoing research and is expected to enhance the representation of water resources in spatially and temporally varying wet and dry conditions. This potential is demonstrated here through the implementation of horizon curves in the simulation of four key dams in the Western US. The operating policy information (i.e., release-availability function) derived in this work could also be explored on its own merits. For example, one could compare the lower limits of release across all time periods and dams to explore variation in environmental releases. Deriving new horizon curves for different periods of history may reveal changing preferences—such as points in time where environmental releases have increased, or the first introduction of forecast use in decision making.

**Acknowledgements**

This research was supported by the U.S. Department of Energy, Office of Science, as part of research in Multi-Sector Dynamics, Earth and Environmental System Modeling Program. This work was authored by the Pacific Northwest National Laboratory, managed by Battelle under contract DE-AC05-76RL01830 for the U.S. Department of Energy (DOE). The views expressed in the article do not necessarily represent the views of the DOE or the U.S. Government. We thank Andy Wood (NCAR), Jeff Arnold (USACE), Ken Nowak (USBR), and Levi Brekke (USBR), and two anonymous reviewers for helpful insight and discussions.

**Code/Data availability**

Data used in this paper are freely available through US Bureau of Reclamation (https://usbr.gov/pn/hydromet/, https://water.usbr.gov), US Army Corps of Engineers (via Duke University, https://nicholasinstitute.duke.edu/reservoir-data/), California Data Exchange Center (https://cdec.water.ca.gov), Texas Water Development Board (https://waterdatafortexas.org/reservoirs/statewide), and US Geological Survey (https://waterdata.usgs.gov/). All code and calculations used to identify weekly forecast horizons are available at https://github.com/IMMM-SFA/horizon.

**Author contribution**

ST and NV designed the study. ST developed the horizon curve derivation approach, wrote the horizon software, and ran numerical experiments. WX designed and executed the Random Forest classification. NV supervised the study. ST prepared the manuscript with contributions from all co-authors.

**Competing interests**

The authors declare that they have no conflict of interest.

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

**Figure 1 – Example of derivation of horizon curve for a given dam using piecewise linear functions fitted to release (*r*) availability (*a*) scatters. Best fit horizons (based on coefficient of determination) for each water week (i.e., each row of the release-availability plots above) are combined to create the horizon.**

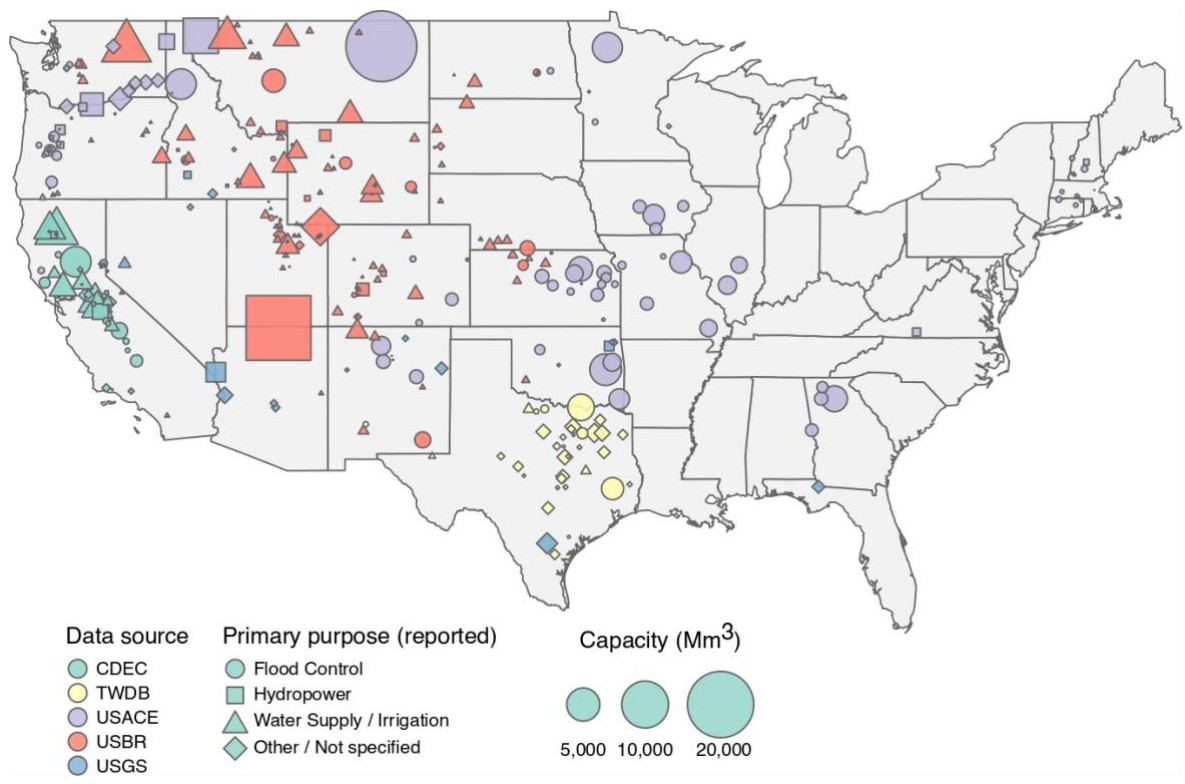

**Figure 2 – Dams included in forecast use signature analysis (n = 316). Data sources are California Data Exchange (CDEC), Texas Water Development Board (TWDB), US Corps of Engineers (USACE), US Bureau of Reclamation (USBR) and US Geological Survey (USGS).**

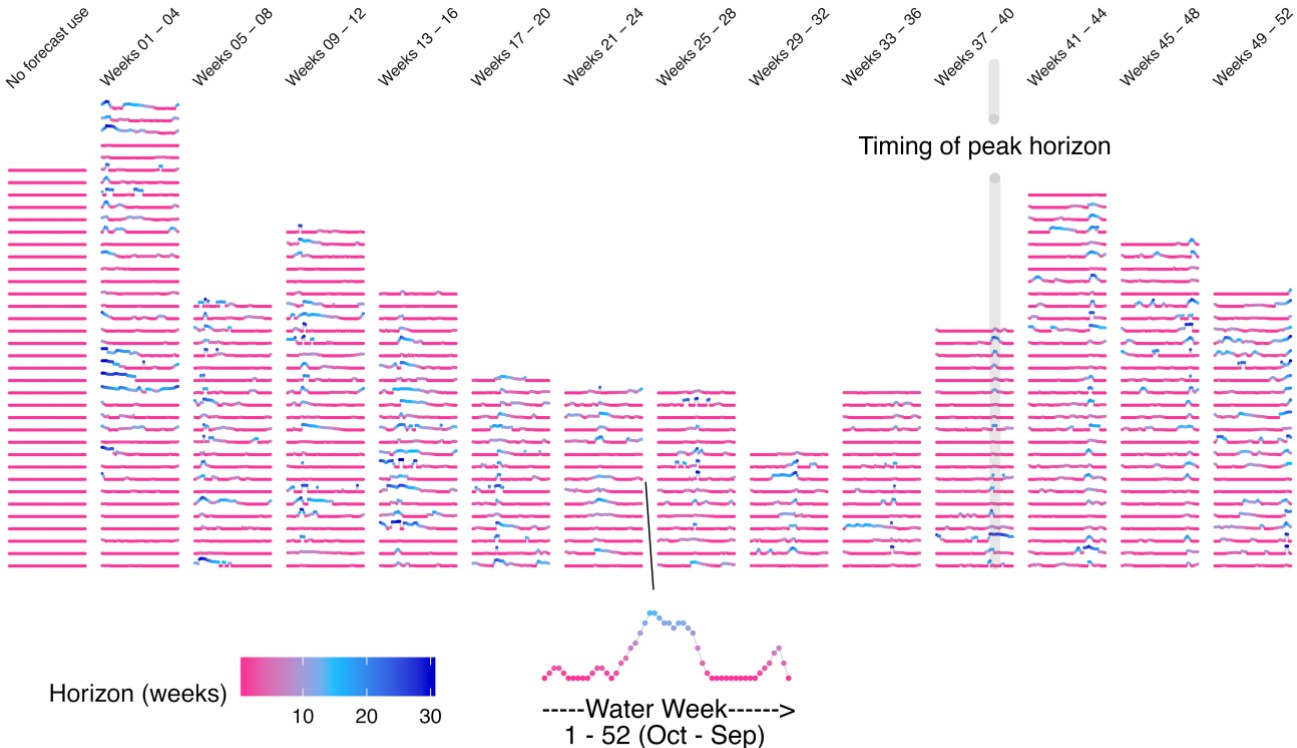

**Figure 3 – Horizon curves for 316 dams, binned according to timing of peak horizon (i.e., the week of the water year where the longest-range foresight horizon is detected). Each signature specifies the inferred operational horizon from water week 1 (week commencing 1st October, at the left of forecast use signature) to 52 (week commencing September 24th, at the very right of the forecast use signature).**

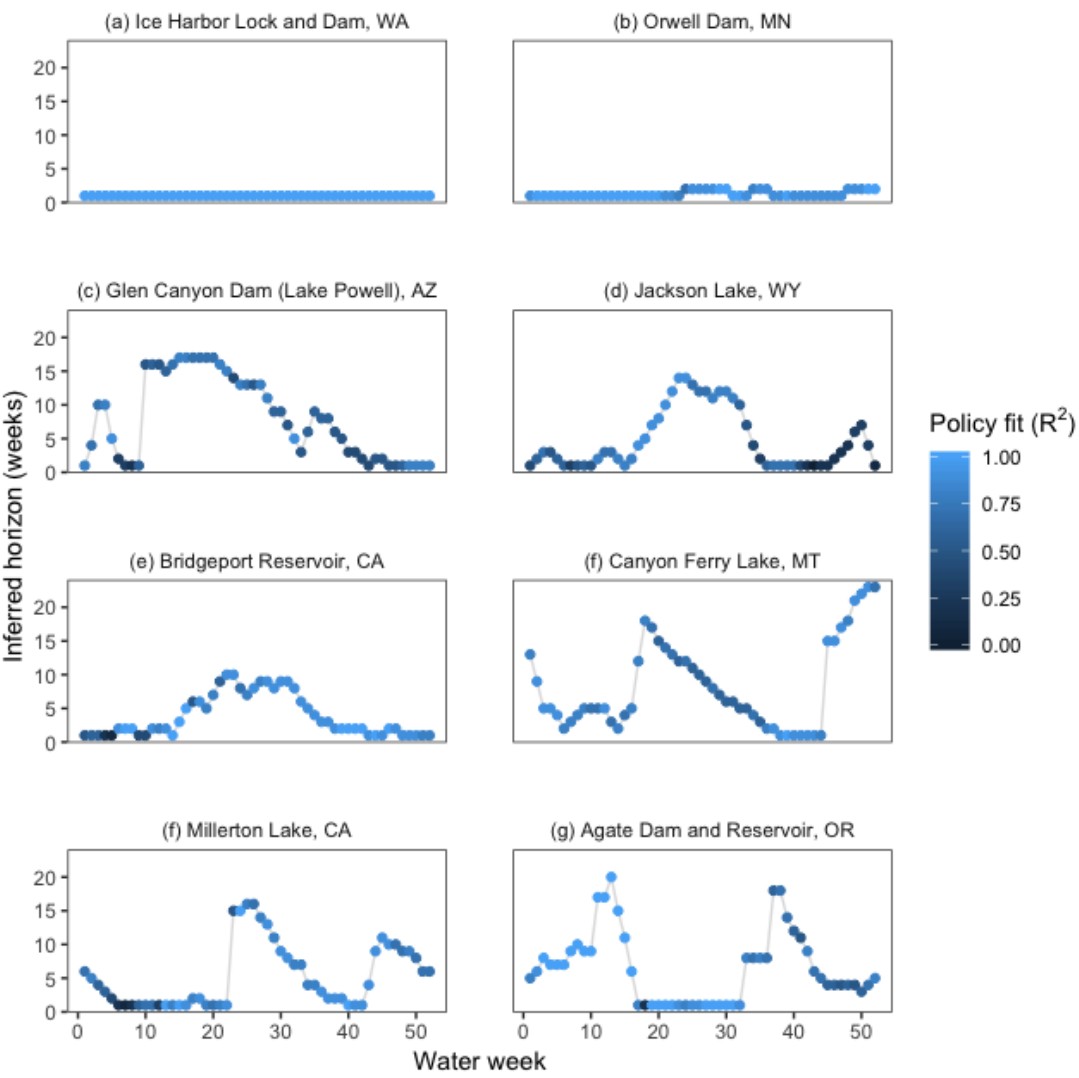

**Figure 4 – Inflow forecast use signature examples for eight dams located throughout Western United States. The policy fit refers to the coefficient of determination ($R_2$) of the release-availability relationship for the best-fit horizons.**

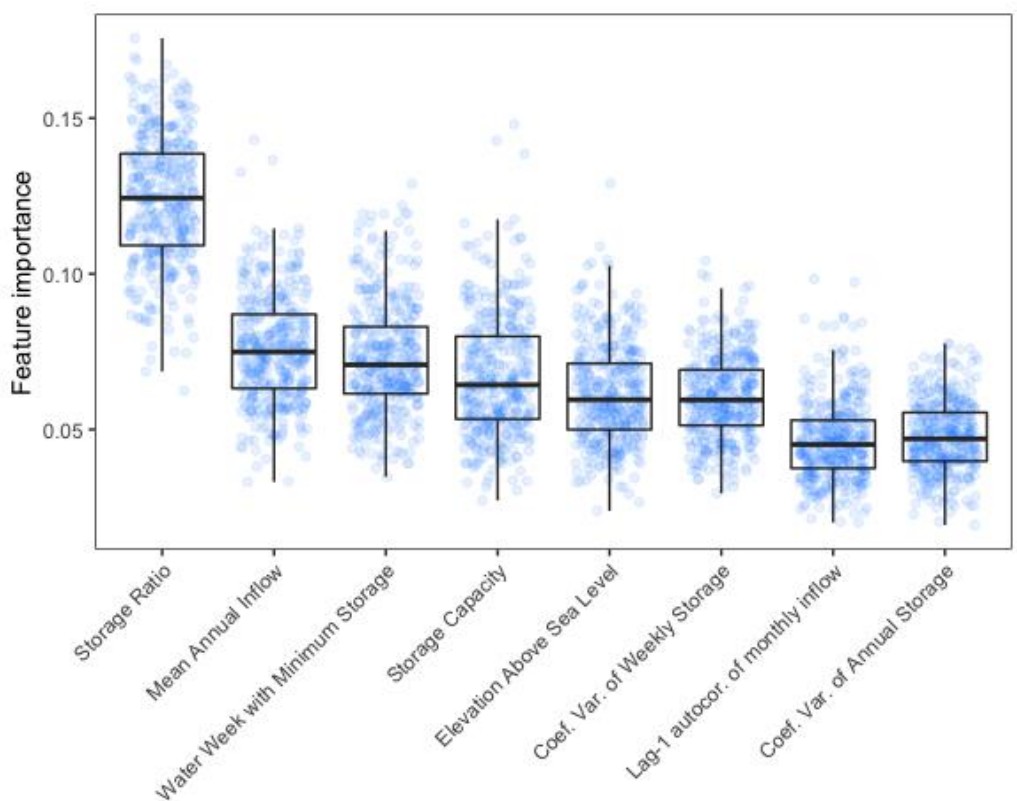

**Figure 5 – Distribution of feature importance across 400 random forests (eight features with highest median importance shown). The distribution is created by bootstrapping the random forest classification model with resampled training and test data. Boxplots give median and interquartile range.**

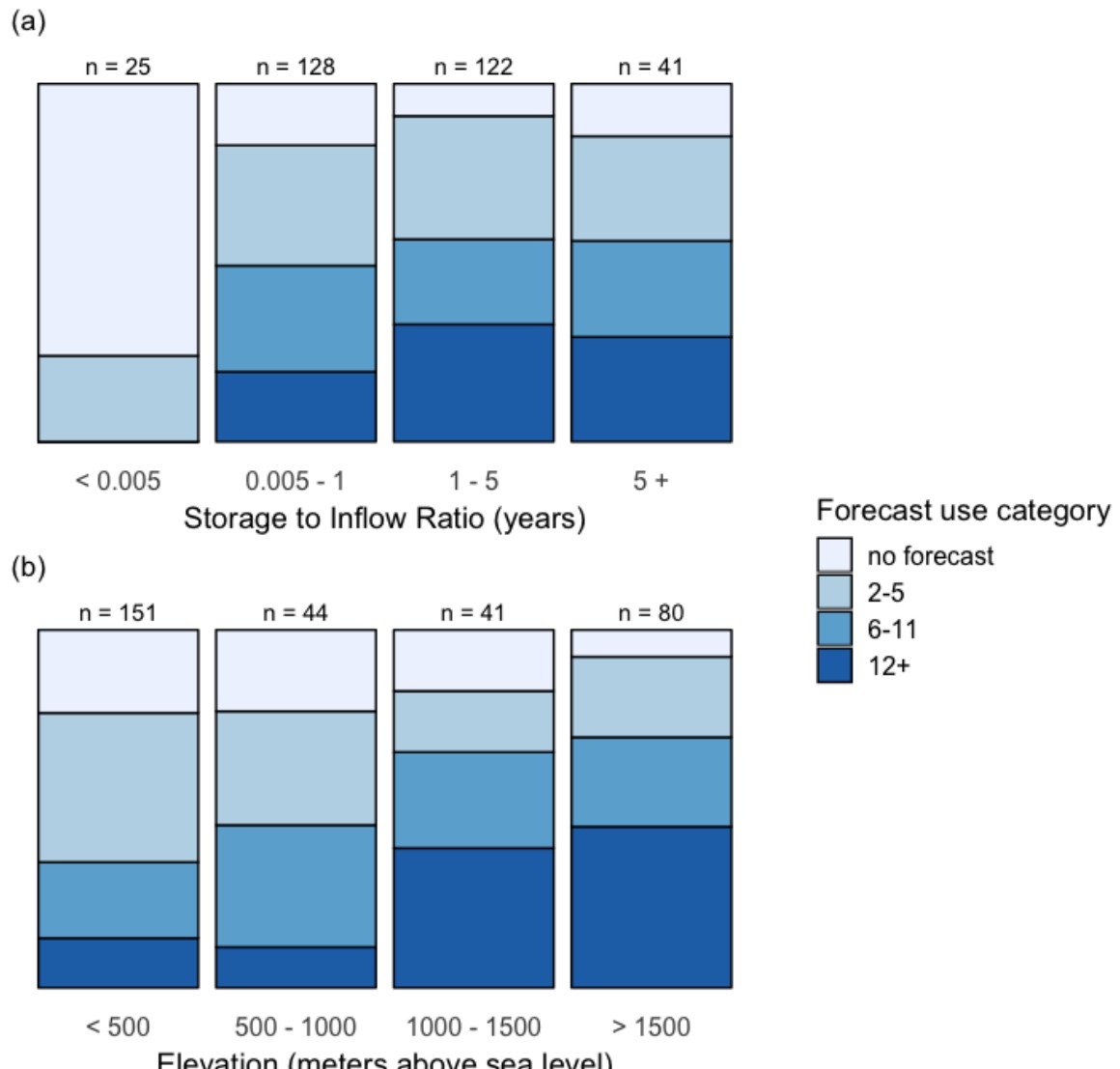

**Figure 6 – Stacked (100%) bars showing distribution of dams by detected forecast horizon within categories of (a) storage ratio and (b) elevation. Forecast use categories are: no forecast, 2 – 5 weeks ahead horizon, 6 – 11 weeks ahead horizon, and 12 weeks or greater horizon. Maximum detected horizon assumes that the horizon is detected in the forecast use signature for at least three consecutive weeks with policy fit ($R^2$) exceeding 0.5 in each week.**

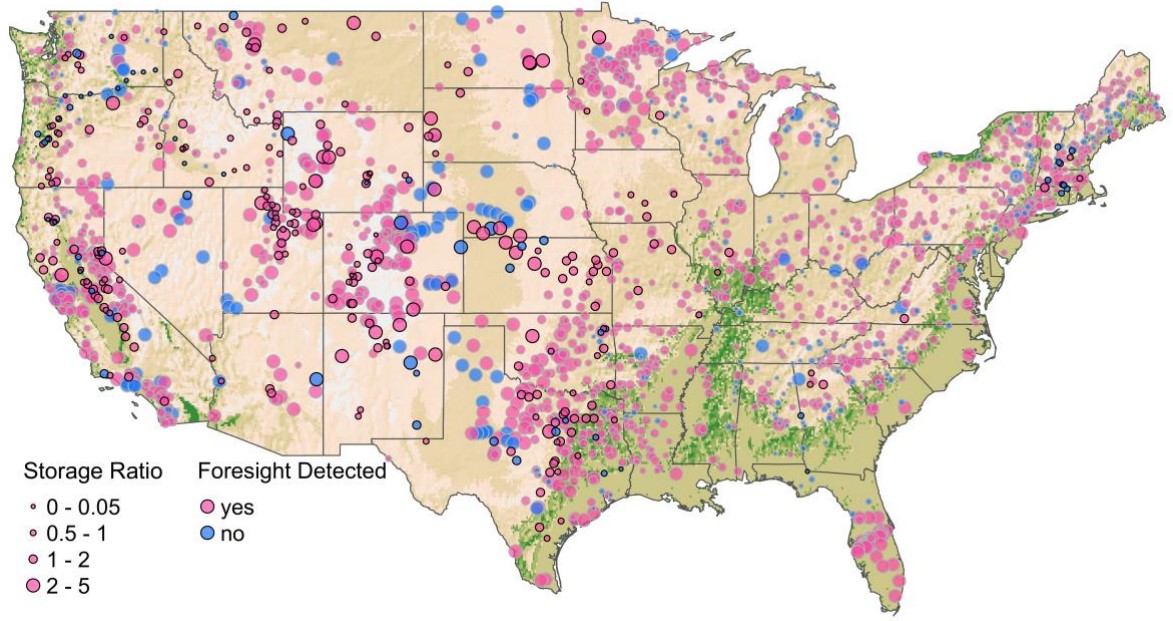

**Figure 7 – Foresight-use for 1942 CONUS dams and reservoirs, based on forecast-use signatures (316 dams – black outlined circles) and out-of-sample, extrapolated estimates (gray outlined circles). Storage ratio (split into four categories) is storage capacity divided by the annual average reservoir inflow. Background shading gives land elevation.**


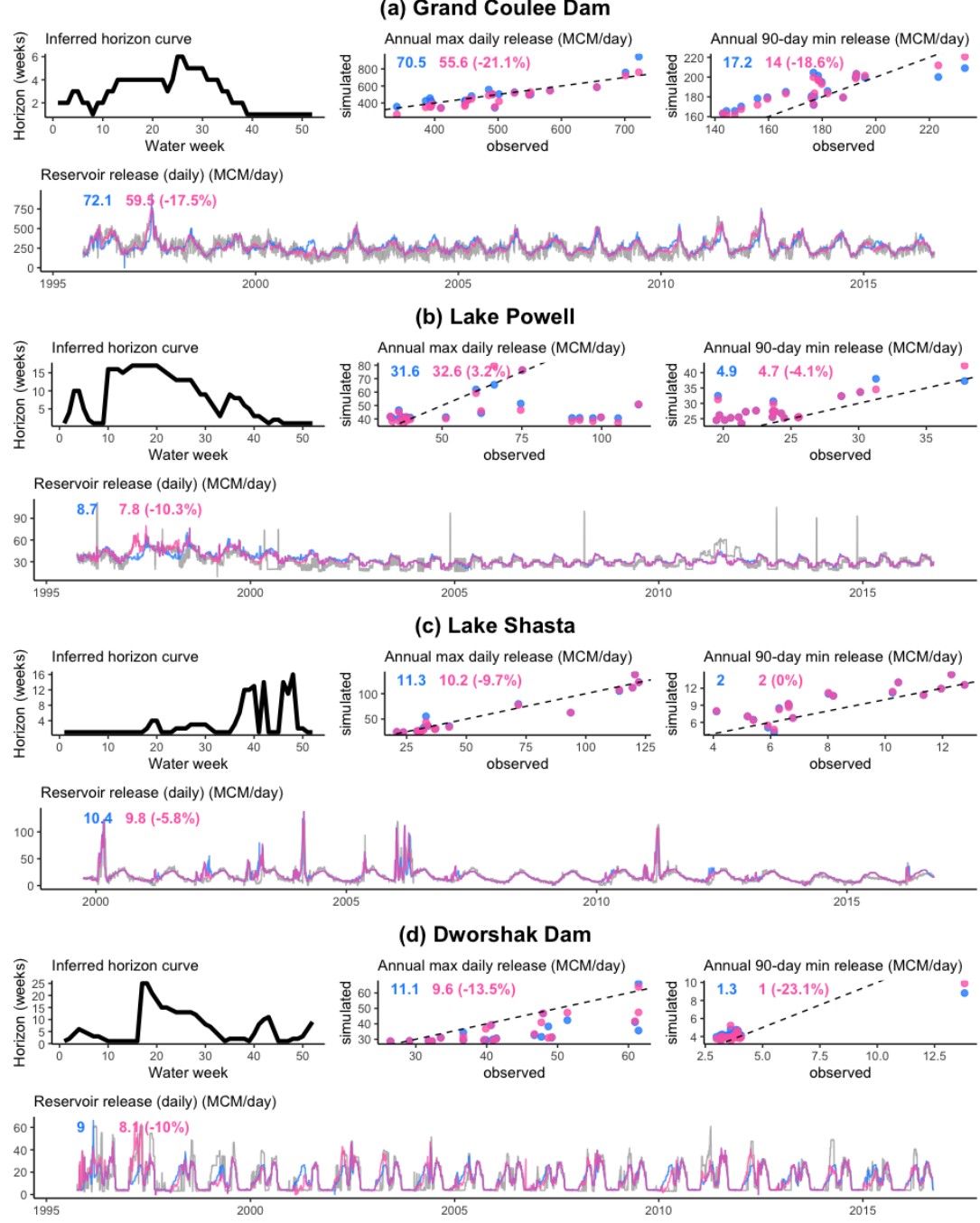

**Figure 8 – Simulation performance improvements with horizon curves adopted. Blue represents forecast excluded; pink represents forecast included. Results given for four large storage dams, showing the inferred horizon curve, scatter plots for annual maxima and annual minima (90-day average) releases (representing performance during flood and drought conditions respectively) and the daily release time series. Numbers inside plot panels give RMSE scores relative to observation (% difference with horizon curve in parentheses).**

**Table 1 – Performance metrics for observed versus simulated daily release for two policies: "current week" (CW) neglects forecasts and instead uses piecewise release-availability functions trained on availability = storage + current inflow; "horizon curve" (HC) adopts the horizon curve and associated piecewise release-availability functions. Metrics assessed are RMSE of daily releases, RMSE of annual maxima of the daily releases, RMSE of annual 90-day minimum series, and Box-Cox transformed RMSE (TRMSE).**

| | Grand Coulee | | Lake Powell | | Shasta | | Dworshak | |
|---|---|---|---|---|---|---|---|---|
| | CW | HC | CW | HC | CW | HC | CW | HC |
| $RMSE_{r\_daily}$ | 72.1 | 59.5 *(-17.5%)* | 8.7 | 7.8 *(-10.3%)* | 10.4 | 9.8 *(-5.8%)* | 9.0 | 8.1 *(-10.0%)* |
| $RMSE_{r\_ann\_max}$ | 70.5 | 55.6 *(-21.1%)* | 31.6 | 32.6 *(+3.2%)* | 11.3 | 10.2 *(-9.7%)* | 11.1 | 9.6 *(-13.5%)* |
| $RMSE_{r\_ann\_90dmin}$ | 17.2 | 14.0 *(-18.6%)* | 4.9 | 4.7 *(-4.1%)* | 2.0 | 2.0 *(-)* | 1.3 | 1.0 *(-23.1%)* |
| $TRMSE_r$ | 1.58 | 1.28 *(-15.4%)* | 0.68 | 0.62 *(-8.7%)* | 1.12 | 1.06 *(-5.4%)* | 1.25 | 1.12 *(-10.5%)* |

**Appendix**

Table A1 – Sensitivity of results to change in number of consecutive weeks of horizon detected required to label a dam as having a significant horizon curve. The 26 features determined for each dam and included in this analysis are: dam elevation above sea level, dam purpose (a single categorical variable for primary purpose as well as seven Boolean variables indicating whether dam is used for water supply, irrigation, flood control, recreation, hydropower, ecological provision, and navigation
respectively), dam latitude, dam longitude, reservoir storage capacity, storage ratio (i.e., storage to annual inflow ratio), coefficient of variation of storage (annual and weekly time series), mean annual inflow, coefficient of variation of inflow (annual and weekly) lag-1 autocorrelation of inflow (annual, quarterly, monthly, and weekly), average week of water year when minimum inflow occurs, average week of water year when maximum inflow occurs, number of dams upstream, total capacity of dams upstream.

| Consecutive weeks of horizon detected required to label the horizon curve "significant" | Observed number of dams with significant horizon curve | Predicted number of dams >10 MCM with significant horizon curve | Top five predictive features for whether dam's horizon curve is significant, ordered by mean importance in random forest classification |
|---|---|---|---|
| 2 | 277 (88%) | 1677 ± 6 (87%) | *Storage ratio (0.15)* |
| | | | *Elevation (0.09)* |
| | | | Mean ann. inflow (0.08) |
| | | | CV of weekly storage (0.06) |
| | | | Week of minimum flow (0.06) |
| 3 (in study) | 258 (82%) | 1553 ± 50 (81%) | *Storage ratio (0.12)* |
| | | | Mean ann. inflow (0.08) |
| | | | Week of minimum flow (0.06) |
| | | | Storage capacity (0.07) |
| | | | *Elevation (0.06)* |
| 4 | 223 (71%) | 1219 ± 40 (63%) | *Storage ratio (0.09)* |
| | | | Lag-1 ACF of mon. inflow (0.08) |
| | | | CV of annual storage (0.07) |
| | | | *Elevation (0.06)* |
| | | | CV of weekly storage (0.06) |

**Definitions of feature performance scores:**

In validation of the classification model, True Positives (TP) is the number of correctly predicted TRUE values (i.e., forecast detected) while False Positives (FP) is the number of incorrectly predicted TRUE values (i.e., model predicts "forecast detected = TRUE" when it should be FALSE). Similarly, True Negatives (TN) is the number of correctly predicted FALSE values and False Negatives (FN) is the number of incorrectly predicted FALSE values.

Precision = TP / (TP + FP)

Recall = TP / (TP + FN)

F1 = (2 × Precision × Recall) / (Precision + Recall)