# Peer review of "Inferred inflow forecast horizons guiding reservoir release decisions across the United States"

_Hydrology and Earth System Sciences, 2019_

## Referee Comment (RC1) · Anonymous Referee #1 · 4 Nov 2019

This paper contributes 1) the novel notion of "horizon curve", i.e., a week-by-week assessment of the forecast horizon that is most relevant to release decision during a certain week, and 2) a methodology to derive this curve, with an application to a number of CONUS dams. This methodology is complemented by a random forest analysis to link results with dam characteristics, and with an illustration that integrating horizon curve within a release rule can improve the fit with observations.

The idea is creative and very timely, as modelers are working to improve the representation of reservoir release rules within hydrological models. Its strength lies in trying to summarize complex forecast processes, based on disparate and often site-specific

information (as authors discuss in the paper), with a single vector mapping the forecast horizon as a function of time of year. Authors are able to substantiate their results with supplemental analysis, and to interpret the results for a range of different situations (this also demonstrates that the methodology works well enough to adapt the results to these different configurations). The paper is also well-written. Overall, this fits well within the remit of HESS both in terms of scope and quality.

There are however, aspects that authors should address to increase the quality and rigor of this publication. Comments presented hereafter are in no particular order.

Authors correctly identify (line 113-136) that stationarity in operations and forecast availability is a necessary assumption (and / or limitation) here, but they don't actually use the word or relate it to hydrological (non-)stationarity. I would advise authors to move that paragraph to the discussion and relate their assumption to climatic, operational and forecast stationarities: all of them refer to sources of information that reservoir operators may rely on and that may (or do) vary over time.

Line 59: please replace "hydrology" by "hydrological"

Line 93: mention that October 1st is the start of the hydrological year.

Line 106: it would be good to tell right there (maybe with an equation) how exactly fit is computed, and justify choice of formula.

Section 2.4 and line 271: it would be useful to precisely define F1 score, precision and recall.

Discussion and Figure 8: it is unclear what the release rule is and how exactly the horizon curves have been incorporated to it. Improved results from integrating horizon curves within the release rule representation are the exclamation point to this paper (and authors are right to mention in the abstract), so the appropriate details should be given. This should be presented in the methodology (Section 2) as an example of how horizon curves can be implemented in practice. Then results should be described at

the end of Section 3 instead of in the discussion. Then lines 21-23: the statement is far too general and assertive. Authors only show that integrating forecast information CAN improve performance, using an unspecified release rule at selected sites.

With this Figure 8 description of how accounting for the horizon curve can improve the representation of reservoir operations, authors may be missing an opportunity by only using the RMSE of release as a performance indicator. This is a general comment that could be better qualified by focusing on very high or very low simulated flows. For instance, they could use goodness-of-fit indicators especially designed to highlight the quality of the fit for high or low flows (see van Werkhoven et al., Advances in Water Resources 32 (2009) 1154–1169). Alternatively, they could define a reservoir's spill as the outlfow beyond the maximal quantity of water that can go through hydropower turbine in a day, and look at spill RMSE. Integrating forecast would likely have a significant effect on spills, e.g., in a case like Lake Powell.

A final, general comment is that in the absence of pointers on what the forecast information is, the forecast information might well be the expected average inflows conditions – involving no actual forecast at all. This should be clarified.

---

## Author Comment (AC1) · 7 Nov 2019

Thank you for your review and for your constructive suggestions for improving the article. We agree with all of the suggestions and will respond fully in due course. We wish to make an immediate short comment on the very last point raised, so that the issue can be clarified for remaining reviewers.

The comment was: "A final, general comment is that in the absence of pointers on what the forecast information is, the forecast information might well be the expected average inflows conditions – involving no actual forecast at all. This should be clarified."

[Figure]

Please be assured that we have constructed the method so that expected average inflow conditions will not be detected as forecast information. From Line 108:

"Importantly, because the algorithm is computed for individual weeks, it removes the effects of operational decisions driven by typical water availability conditions throughout the year. For example, simple knowledge that springtime typically brings high flows would not register as foresight in this procedure. Foresight detected must result from some knowledge as to how incoming flows differ from usual for that time of year."

This was perhaps an uninformative way of making the point. Instead consider typical release rules for a reservoir. Those rules specify how much water to release as a function of storage for different times of year. There's no look ahead horizon (release is only a function of current water availability), and yet the policy is still one that releases water to deal with seasonality in inflow conditions (since the relationship between release and water availability varies throughout the year). If it's ahead of flood season, releases will be high to draw down the reservoir to make a flood pool, and so forth. Thus, when examining the decisions of a given week of the year, we would find a close relationship between release and current water availability (as per the policy) and would expect the policy fit to deteriorate by extending the candidate horizon (since the time of year, rather than an explicit forecast of water, is informing the decision). In this way, releases designed for average seasonality of inflow would not register as foresight in our procedure (we do it one week at a time). We have reflected that this can be difficult for a reader to digest, so we'll provide an improved explanation, diagram, mathematical proof, or demonstrative simulation in our revision if given the opportunity.

Thanks again for your very helpful review.

Sean Turner (lead author) PNNL

---

## Referee Comment (RC2) · Anonymous Referee #2 · 18 Nov 2019

Review of "Inferred inflow forecast horizons guiding reservoir release decisions across the United States" by Turner et al.

In this submitted manuscript, the authors applied a conceptual method to analyze 316 dams and reservoirs in the U.S. with respect to the roles of forecast information in driving the discharge operations. Using the proposed "horizon curves", authors specifically analyzed the relationship of forecast information and operation for four key dams, in order to test whether the proposed method could improve the modeling capability of large-scale hydrological studies with the interference of reservoirs and dams. The study was originated from the fact that water managers nowadays will rely more and

more on hydrological forecasts as the information of forecast range, resolution, and accuracy have been improved by various methods. However, it is still unknown how important and how influential a forecast could essentially improve the reservoirs and dams operations. One of the contributions of this study is to analyze a large number of dams in the United States based on limited reservoir operation data. In addition, a new concept of "horizon curves" has been proposed by authors. Using the proposed method, this study also tries to answer the question of "how and when do forecasts applied in the field of reservoir operations". In general, the scope of the submitted manuscript is indeed very interesting, and the main contribution & novelty lies in the invention of the concept of "horizontal curve" for reservoir operations. However, the reviewer thinks there are a few key assumptions are questionable when authors develop the "horizon curve" method. Those assumptions are subject to verification and further investigation. In addition, the reviewer also finds the organization of the manuscript is confusing, and few paragraphs in the methodology section are hard to follow due to missing steps or information. Some major issues are listed as follow:

1. Methodology Justification (Line 85-86) The proposed approach is based on the assumption that 1) "the future observed inflows (perfect forecast) may act as a proxy for the actual forecast available to the operator at the time of deciding how much water to release". However, in reality, reservoir operators never trust a single forecast, at least the forecast uncertainty needs to be considered when making any release decision. More importantly, most of the releases are pre-defined by the reservoir "rule curves" with limited influences from the forecasts. Regardless of the forecasts in different ranges and accuracy, reservoir operators always and have to releases a certain amount of water at a certain time following the "rule curves". Therefore, the reviewer is unclear how the forecast information based on the "horizon curves" could actually interact with the existing operating rules. The manuscript seems to omit this linkage between forecasts and the hard rules reservoir operators must follow. The rule curves are even more important in terms of mid to long-term operations, which is the same time range this study has been focused on. Different reservoirs have different rule

curves, and it would vary from one reservoir to another. The proposed method of "horizon curves" seems to be a universal approach for any reservoirs. Reviewer is wondering the applicability of the "horizon curves" as each reservoir will have different settings and "rules" to follow as regulated by USACE or relevant water agencies. How does the proposed "horizon curves" could address different reservoir regulations and functions, such as hydropower reservoir vs. flood control reservoir vs. water supply reservoir vs. environmental demands?

2. The steps of horizon curve method (Line 80-120) The reviewer cannot fully understand the whole process of deriving the horizon curve. For example, in line 98, the authors mentioned the "release-availability" function for the first time in the manuscript and then briefly explained the definition of "availability a". How do the authors define "release-availability" here? How did authors construct the functions of "release-availability"? The possible releases must be from the existing operating rules to prevent overtopping and dead-pool of reservoir storage. Where do authors obtain such information in a national scale? This is a term again not commonly used by water managers, and more explanations would be needed. The authors also wrote, "The inferred policy function fitted to these data is a piecewise linear model with a single breakpoint" in line 99 and the reviewer is wondering what does the "policy function" here refer to? And what do "these data" referred to as? In general, the reviewer thinks this section is hard to follow given lots of non-common terms were used. Those wordings may make sense to authors themselves, however, it is not apparent to water managers and operators. Please re-check some of the literature and especially reservoir operation reports to further explain how the proposed method is constructed in detail. A flowchart or additional figure may be added to explain the steps of creating this horizon curves.

3. The use of Random Forest Classifier (Line 160 - 170): There are few nested issues about the description on the use of Random Forest Classifier, experiment setting, and how will these experiment settings lead to a conclusion related to the forecast information and reservoir operation. First, authors should point out what are the "features"

and "target" when using Random Forest Classifier. The authors mentioned there are 26 features without a tabular form to let readers know what those are. In addition, the "target" used in RF is still unclear. Did the authors intend to figure out which feature has the most important influences on release decisions? Or did the authors intend to classify reservoirs according to their correlation between discharge and inflow forecasts? And how was this realized in RF? The description here reads very short and is not comprehensive. Reviewer is confused about what has been classified based on what inputs, as well as how this experiment setting would lead to a certain conclusion. At least a few additional paragraphs would be necessary to explain the experiments here.

4. data segmentation (Line 160 - 170): Since the methodology used here is Random Forests as one of the machine learning tools, the reviewer is wondering whether there is an overfitting issue? Traditionally, the data should be split into training, validation and test periods to verify there is no overfitting. However, authors only did a training and a test without validation. It is likely the model is overfitted and more experiments on different folds are necessary to justify the proper use of random forests.

5. Gini index Line 305: Can the authors define what is the "Gini impurity of the tree"? Some examples and references using this index would be helpful.

―――――――――――――――――

---

## Author Comment (AC2) · 10 Dec 2019

**Referee Comment:** This paper contributes 1) the novel notion of "horizon curve", i.e., a week-by-week assessment of the forecast horizon that is most relevant to release decision during a certain week, and 2) a methodology to derive this curve, with an application to a number of CONUS dams. This methodology is complemented by a random forest analysis to link results with dam characteristics, and with an illustration that integrating horizon curve within a release rule can improve the fit with observations. The idea is creative and very timely, as modelers are working to improve the representation of reservoir release rules within hydrological models. Its strength lies in

trying to summarize complex forecast processes, based on disparate and often site-specific information (as authors discuss in the paper), with a single vector mapping the forecast horizon as a function of time of year. Authors are able to substantiate their results with supplemental analysis, and to interpret the results for a range of different situations (this also demonstrates that the methodology works well enough to adapt the results to these different configurations). The paper is also well-written. Overall, this fits well within the remit of HESS both in terms of scope and quality.

**Author Response:** Thank you very much for your time and for your thoughtful and constructive review.

**Proposed changes to manuscript:** N/A

**Referee Comment:** Authors correctly identify (line 113-136) that stationarity in operations and forecast availability is a necessary assumption (and / or limitation) here, but they don't actually use the word or relate it to hydrological (non-)stationarity. I would advise authors to move that paragraph to the discussion and relate their assumption to climatic, operational and forecast stationarities: all of them refer to sources of information that reservoir operators may rely on and that may (or do) vary over time.

**Author Response:** We agree that the term stationarity would help readers better understand this limitation and that the issue should, in part, be related to hydrological non-stationarity (although we suspect the most important driver of non-stationarity in operating rules would be introduction of environmental flow regulations). Another interesting source of hydrological non-stationarity would be a change in operations at an upstream dam (as opposed to climate-driven).

**Proposed changes to manuscript:** We propose to expand the discussion of non-stationarity as suggested by the reviewer.

**Referee Comment:** Discussion and Figure 8: it is unclear what the release rule is and how exactly the horizon curves have been incorporated to it. Improved results from

integrating horizon curves within the release rule representation are the exclamation point to this paper (and authors are right to mention in the abstract), so the appropriate details should be given. This should be presented in the methodology (Section 2) as an example of how horizon curves can be implemented in practice. Then results should be described at the end of Section 3 instead of in the discussion. Then lines 21-23: the statement is far too general and assertive. Authors only show that integrating forecast information CAN improve performance, using an unspecified release rule at selected sites.

**Author Response:** Thank you for highlighting this missing piece of information. The release is computed from the set of optimal piecewise functions (one for each week) that result in the closest fit when generating the horizon curve for each dam. For example, if the best piecewise function for water week X uses a 7-week ahead horizon, then that function will be used to determine the release in re-simulation for week X. We agree that we should expand the method to clarify this for readers.

**Proposed changes to manuscript:** We will follow the reviewer's suggestions to: (1) expand the method section to describe how the release rule is generated as part of the horizon curve derivation; (2) move the associated figure into results, and (3) change the text in the abstract so that our claim (horizon curve improves simulation performance) reflects the limited nature of the simulation analysis involving eight reservoirs.

**Referee Comment:** With this Figure 8 description of how accounting for the horizon curve can improve the representation of reservoir operations, authors may be missing an opportunity by only using the RMSE of release as a performance indicator. This is a general comment that could be better qualified by focusing on very high or very low simulated flows. For instance, they could use goodness-of-fit indicators especially designed to highlight the quality of the fit for high or low flows (see van Werkhoven et al., Advances in Water Resources 32 (2009) 1154–1169). Alternatively, they could define a reservoir's spill as the outflow beyond the maximal quantity of water that can go through hydropower turbine in a day, and look at spill RMSE. Integrating forecast

would likely have a significant effect on spills, e.g., in a case like Lake Powell.

**Author Response:** We had actually looked at some other metrics and decided to present just RMSE for simplicity (we reach similar conclusions with NSE and KGE). We agree that the metrics that assess performance of low flows would be a useful addition.

**Proposed changes to manuscript:** We will tabulate performances across two or three different performance metrics for each reservoir, and also show results for goodness of fit of the storage and spill time series where observations permit.

**Referee Comment:** A final, general comment is that in the absence of pointers on what the forecast information is, the forecast information might well be the expected average inflows conditions – involving no actual forecast at all. This should be clarified.

**Author Response:** Please see the separate comment, posted immediately following this review, that addresses this comment exclusively.

**Proposed changes to manuscript:** We will re-write this section of the manuscript to demonstrate clearly that our procedure avoids detecting foresight in operations as a result of expected average inflows.

**Referee Comment:** - Line 59: please replace "hydrology" by "hydrological" - Line 93: mention that October 1st is the start of the hydrological year. - Line 106: it would be good to tell right there (maybe with an equation) how exactly fit is computed, and justify choice of formula. - Section 2.4 and line 271: it would be useful to precisely define F1 score, precision and recall.

**Author Response:** We agree with all of the minor changes above.

**Proposed changes to manuscript:** We will implement all of these adjustments, corrections, and clarifications.
* * *
486, 2019.

---

## Author Response (AR1)

**We would like to thank both reviewers for their time and constructive feedbacks. Please find below our clarifications and actions taken. Changes are referenced with line numbers in the revised manuscript. These line numbers are the same for the final and "tracked-changes" versions.**

**Reviewer 1**

**Referee Comment:** This paper contributes 1) the novel notion of "horizon curve", i.e., a week-by-week assessment of the forecast horizon that is most relevant to release decision during a certain week, and 2) a methodology to derive this curve, with an application to a number of CONUS dams. This methodology is complemented by a random forest analysis to link results with dam characteristics, and with an illustration that integrating horizon curve within a release rule can improve the fit with observations. The idea is creative and very timely, as modelers are working to improve the representation of reservoir release rules within hydrological models. Its strength lies in trying to summarize complex forecast processes, based on disparate and often site-specific information (as authors discuss in the paper), with a single vector mapping the forecast horizon as a function of time of year. Authors are able to substantiate their results with supplemental analysis, and to interpret the results for a range of different situations (this also demonstrates that the methodology works well enough to adapt the results to these different configurations). The paper is also well-written. Overall, this fits well within the remit of HESS both in terms of scope and quality.
**Author Response:** Thank you for your thoughtful and constructive review.

**Referee Comment:** Authors correctly identify (line 113-136) that stationarity in operations and forecast availability is a necessary assumption (and / or limitation) here, but they don't actually use the word or relate it to hydrological (non-)stationarity. I would advise authors to move that paragraph to the discussion and relate their assumption to climatic, operational and forecast stationarities: all of them refer to sources of information that reservoir operators may rely on and that may (or do) vary over time.
**Author Response:** We agree that the term stationarity would help readers better understand this limitation and that the issue should, in part, be related to hydrological non-stationarity (although we suspect the most important driver of non-stationarity in operating rules would be introduction of environmental flow regulations).
**Changes to manuscript:** We have added some discussion on the idea of non-stationarity [*LINES 351 – 356*].

**Referee Comment:** Discussion and Figure 8: it is unclear what the release rule is and how exactly the horizon curves have been incorporated to it. Improved results from integrating horizon curves within the release rule representation are the exclamation point to this paper (and authors are right to mention in the abstract), so the appropriate details should be given. This should be presented in the methodology (Section 2) as an example of how horizon curves can be implemented in practice. Then results should be described at the end of Section 3 instead of in the discussion. Then lines 21-23: the statement is far too general and assertive. Authors only show that integrating forecast information CAN improve performance, using an unspecified release rule at selected sites.

**Author Response:** Thank you for highlighting this missing piece of information. The release is computed from the set of optimal piecewise functions (one for each week) that result in the closest fit when generating the horizon curve for each dam. For example, if the best piecewise function for water week X uses a 7-week ahead horizon, then that function will be used to determine the release in re-simulation for week X. We agree that we should expand the method to clarify this for readers.

**Changes to manuscript:** (1) we have expanded the method section to describe how the release rule is generated as part of the horizon curve derivation *[LINES 198 – 201]*; (2) move the associated figure into the results section, and (3) change the text in the abstract so that our claim (horizon curve improves simulation performance) reflects the limited nature of the simulation analysis involving eight reservoirs in offline mode *[LINES 21 – 22]*.

**Referee Comment:** With this Figure 8 description of how accounting for the horizon curve can improve the representation of reservoir operations, authors may be missing an opportunity by only using the RMSE of release as a performance indicator. This is a general comment that could be better qualified by focusing on very high or very low simulated flows. For instance, they could use goodness-of-fit indicators especially designed to highlight the quality of the fit for high or low flows (see van Werkhoven et al., Advances in Water Resources 32 (2009) 1154–1169). Alternatively, they could define a reservoir's spill as the outflow beyond the maximal quantity of water that can go through hydropower turbine in a day, and look at spill RMSE. Integrating forecast would likely have a significant effect on spills, e.g., in a case like Lake Powell.

**Author Response:** We had actually looked at some other metrics and decided to present just RMSE for simplicity (we reach similar conclusions with NSE and KGE). We agree that the metrics that assess performance of low flows would be a particularly useful addition.

**Changes to manuscript:** We have tabulated performances across a range of metrics *[LINES 318 – 322]* for each reservoir in a table supporting Figure 8 *[TABLE 1, LINE 575]*.

**Referee Comment:** A final, general comment is that in the absence of pointers on what the forecast information is, the forecast information might well be the expected average inflows conditions – involving no actual forecast at all. This should be clarified.

**Author Response:** We can confirm that the process completely avoids inferring as forecasts any operations tailored to average seasonal inflow conditions. Operations tailored to average inflows are a function of time-of-year and can therefore be intercepted by performing the forecast derivation on individual weeks (see short comment posted in immediate response to your review as well as the rewrite of the method section).

**Changes to manuscript:** We have rewritten this section in the manuscript to demonstrate clearly that our procedure avoids detecting foresight in operations as a result of expected average inflows *[LINES 126 – 137]*.

**Minor comments:**
- Line 59: please replace "hydrology" by "hydrological" *[done – LINE 62]*.
- Line 93: mention that October 1st is the start of the hydrological year *[done – LINE 96]*.
- Line 106: it would be good to tell right there (maybe with an equation) how exactly fit is computed, and justify choice of formula *[done - LINES 109 – 112]*.
- Section 2.4 and line 271: it would be useful to precisely define F1 score, precision and recall *[done - LINES 590 – 596]*.

**Reviewer 2**

**Referee Comment:** In this submitted manuscript, the authors applied a conceptual method to analyze 316 dams and reservoirs in the U.S. with respect to the roles of forecast information in driving the discharge operations. Using the proposed "horizon curves", authors specifically analyzed the relationship of forecast information and operation for four key dams, in order to test whether the proposed method could improve the modeling capability of large-scale hydrological studies with the interference of reservoirs and dams. The study was originated from the fact that water managers nowadays will rely more and more on hydrological forecasts as the information of forecast range, resolution, and accuracy have been improved by various methods. However, it is still unknown how important and how influential a forecast could essentially improve the reservoirs and dams operations. One of the contributions of this study is to analyze a large number of dams in the United States based on limited reservoir operation data. In addition, a new concept of "horizon curves" has been proposed by authors. Using the proposed method, this study also tries to answer the question of "how and when do forecasts applied in the field of reservoir operations". In general, the scope of the submitted manuscript is indeed very interesting, and the main contribution & novelty lies in the invention of the concept of "horizontal curve" for reservoir operations. However, the reviewer thinks there are a few key assumptions are questionable when authors develop the "horizon curve" method. Those assumptions are subject to verification and further investigation. In addition, the reviewer also finds the organization of the manuscript is confusing, and few paragraphs in the methodology section are hard to follow due to missing steps or information.
**Author Response:** Thank you for your constructive review. We are confident that we can address each of your concerns through clearer method description.

**Referee Comment:** Methodology Justification (Line 85-86) The proposed approach is based on the assumption that 1) "the future observed inflows (perfect forecast) may act as a proxy for the actual forecast available to the operator at the time of deciding how much water to release". However, in reality, reservoir operators never trust a single forecast, at least the forecast uncertainty needs to be considered when making any release decision. More importantly, most of the releases are pre-defined by the reservoir "rule curves" with limited influences from the forecasts. Regardless of the forecasts in different ranges and accuracy, reservoir operators always and have to releases a certain amount of water at a certain time following the "rule curves". Therefore, the reviewer is unclear how the forecast information based on the "horizon curves" could actually interact with the existing operating rules. The manuscript seems to omit this linkage between forecasts and the hard rules reservoir operators must follow. The rule curves are even more important in terms of mid to long-term operations, which is the same time range this study has been focused on. Different reservoirs have different rule curves, and it would vary from one reservoir to another. The proposed method of "horizon curves" seems to be a universal approach for any reservoirs. Reviewer is wondering the applicability of the "horizon curves" as each reservoir will have different settings and "rules" to follow as regulated by USACE or relevant water agencies. How does the proposed "horizon curves" could address different reservoir regulations and functions, such as hydropower reservoir vs. flood control reservoir vs. water supply reservoir vs. environmental demands?

**Author Response:** Thank you for this comment as it allows us to clarify the application of horizon curves. We agree that we ought to show how the two-stage release function we propose aligns with rule curves and reservoirs that serve multiple functions. Let's first assume that forecasts are not used at all and that we simply want to use the storage volumes to set the release for any given week of the year (as in archetypal rule curve). Our scheme actually allows us to do this very effectively. The two-stage piecewise model provides the flexibility to represent two situations: (a) the reservoir is above the guide-curve, in which case one would expect the operator to respond by releasing water to draw the reservoir back down (with higher release for higher storage levels—allowed for by the right-hand slope of the piecewise function); and (b) the reservoir is below the guide-curve, in which case the operator would wish to cut back the release significantly to allow the reservoir to refill. For (b), the operator would be constrained by the fact that release cannot be negative and that some environmental flow will normally be required, so we add a constraint to our model so that the slope for (b) must be less than for (a). Assuming the operator works only to meet a guide-curve based on storage levels and time of year, our model is ideal. The breakpoint of the piecewise function on the x-axis (water availability) essentially represents the reservoir guide-curve level (since water availability is just storage volume when there is no inflow forecast). Since these rules are optimized to fit observations for each water week and for each reservoir individually, they do indeed account for the differences across the year and across dams. The model is trained on observed decisions, so is agnostic as to whether the reservoir is used for flood control, irrigation, hydropower, etc. It mimics how water is released in practice, so will in effect capture any of these purposes. We must also recognize that these rules are somewhat flexible in the sense that operators will deviate from them depending on forecast information available. Our scheme is designed to allow us to integrate forecast information in the rule-based system. The difference is that the rule is no longer a function of current storage, but becomes a function of current storage plus the inflow out to various candidate horizons, with the candidate horizon offering the best fit to observed decisions taken as the assumed operational horizon in the inferred horizon curve. We agree that operators are more likely in practice to use a forecast ensemble than a deterministic forecast. Given the large scale of our study and the intended application (ultimately a CONUS-scale hydrological and river-routing model) we require a simple proxy for the forecast information available to the operator. Our results demonstrate that the actual, future inflow serves this purpose well in many cases. **Changes to manuscript:** We have amended the method section to explain how our chosen model is compatible with reservoir guide-curves used in real-world operations for a variety of purposes *[LINES 114 – 125]*. We have also added to the discussion to highlight the limitation that our model is based on the assumption of deterministic forecast-use *[LINES 345 – 346]*.

**Referee Comment:** The steps of horizon curve method (Line 80-120) The reviewer cannot fully understand the whole process of deriving the horizon curve. For example, in line 98, the authors mentioned the "release-availability" function for the first time in the manuscript and then briefly explained the definition of "availability a". How do the authors define "release-availability" here? How did authors construct the functions of "release- availability"? The possible releases must be from the existing operating rules to prevent overtopping and dead-pool of reservoir storage. Where do authors obtain such information in a national scale? This is a term again not commonly used by water managers, and more explanations would be needed. The authors also wrote, "The inferred policy function fitted to these data is a piecewise linear model with a single breakpoint" in line 99 and the reviewer is wondering what does the "policy function" here refer to? And what do "these data" referred to as? In general, the reviewer thinks this section is hard to follow given lots of non-common terms were used. Those wordings may make sense to authors themselves, however, it is not apparent to water managers and operators. Please re-check some of the literature and especially reservoir operation reports to further explain how the proposed method is constructed in detail. A flowchart or additional figure may be added to explain the steps of creating this horizon curves.

**Author Response:** Our model is indeed quite complex. Figure 1 was designed to help the reader in this regard, but on reflection we agree that the explanation would be enhanced with a flow chart outlining the process and data used at each stage. The release-availability function is the key to the whole approach. This is simply a relationship (derived for each water week and defined by a broken linear model with two lines and two slopes) that specifies the water release as a function of water availability. These are constructed using observed records of release, storage, and inflow—so the releases are from existing operations designed to prevent overtopping and dead-pool of reservoir storage. We obtain these records from five sources: US Army Corps of Engineers, US Bureau of Reclamation, US Geological Survey, California Data Exchange, and Texas Water Development Board. These are listed in the experimental setup of the manuscript. We used "policy function" and "release-availability" function interchangeably, which we agree is confusing for readers and should be amended.

**Changes to manuscript:** We have defined the release-availability function more clearly *[LINES 100 – 101; 103 – 104; 109 – 112]* (keeping terminology consistent throughout) and have replaced Figure 1 with a flow diagram to guide the reader through process of deriving the horizon curve in an example case *[PAGE 19]*.

**Referee Comment:** The use of Random Forest Classifier (Line 160 - 170): There are few nested issues about the description on the use of Random Forest Classifier, experiment setting, and how will these experiment settings lead to a conclusion related to the forecast information and reservoir operation. First, authors should point out what are the "features" and "target" when using Random Forest Classifier. The authors mentioned there are 26 features without a tabular form to let readers know what those are. In addition, the "target" used in RF is still unclear. Did the authors intend to figure out which feature has the most important influences on release decisions? Or did the authors intend to classify reservoirs according to their correlation between discharge and inflow fore- casts? And how was this realized in RF? The description here reads very short and is not comprehensive. Reviewer is confused about what has been classified based on what inputs, as well as how this experiment setting would lead to a certain conclusion. At least a few additional paragraphs would be necessary to explain the experiments here.

**Author Response:** Thank you. We did indeed define the target, but we referred to it as the "response variable" in the method ("*The response variable is a Boolean of whether there is evidence for significant forecast contribution or not.*") In other words, the target is a simple TRUE/FALSE for whether there is evidence for forecast use in the horizon curve of each dam. The point of the RF analysis is thus to understand if there are features of dams (storage capacity, etc) that would be associated with detected forecast use. We agree that terminology should be made consistent and we expect that with the target clarified this part of the study will be much easier to follow.

**Changes to manuscript:** We have listed all 26 input features of the RF classification *[TABLE A1]* and have adjusted the text for consistency throughout method and results *[LINE 164]*. We have also provided additional detail on the classification within the Method secftion *[LINES 162 - 173]*.

**Referee Comment:** data segmentation (Line 160 - 170): Since the methodology used here is Random Forests as one of the machine learning tools, the reviewer is wondering whether there is an overfitting issue? Traditionally, the data should be split into training, validation and test periods to verify there is no overfitting. However, authors only did a training and a test without validation. It is likely the model is overfitted and more experiments on different folds are necessary to justify the proper use of random forests.

**Author Response:** The traditional machine learning methodology uses training data to train a model, uses validation data to tune model hyperparameters, and uses testing data to evaluate model performance. We agree that if a sole testing set is used for both hyperparameter tuning and model evaluation, the model may be biased and prone to over-fitting. But in our case, the only two hyperparameters tuned (number of trees and maximum tree depth) are not adjusted to achieve best testing score, but instead determined in reflection of our small dataset: maximum tree depth is limited to 3 levels to reduce model complexity, and we choose a considerably large number of trees of 1000 to reduces the model variance and hence reduce overfitting (the larger the number of trees, the lower the ensemble model variance). Additionally, we repeated the experiments 200 times with different splits, each time train the model from scratch on training data and evaluated the model performance only on unseen testing data. We are therefore confident the model is not overfitted.

**Changes to manuscript:** We have added the additional details of the Random Forest experiment design into the draft (including measures taken to avoid overfitting) *[LINES 177 – 180]*.

**Referee Comment:** Gini index Line 305: Can the authors define what is the "Gini impurity of the tree"? Some examples and references using this index would be helpful.

**Author Response:** Thanks for the request for clarification. Gini impurity, like entropy, describes the heterogeneous state of a system and is formally calculated as $\sum_{i=0,1} P_i * (1 - P_i)$ in a binary classification tree node, where $P_i$ is the fraction of samples that belongs to class $i$ in this particular node. The feature importance score is calculated as the percentage of Gini impurity decrease because of the particular feature, averaged across the forest. The sum of all feature importance scores equal to 1.

**Changes to manuscript:** We have added new text to explain Gini impurity (including the equation) and how it is used to calculate feature importance *[LINES 181 - 187]*.

[revised manuscript text omitted]

---

## Author Response (AR2)

Editor summary:

We have received one re-review from Reviewer 1, and the second reviewer declined to re-review, but was mostly positive about the manuscript in their earlier review. Reviewer 1 finds that most of their comments have been well addressed and suggests a minor technical correction. In their earlier review, Reviewer 2 made a number of suggestions, including clarification of the methodology. I find the proposed changes in the revised manuscript address these comments well; the terminology is more consistent and the model approach is easier to understand than before. Therefore, on the basis of these two reviews and my own reading of the revised manuscript, I am happy to recommend publication of this interesting work.

*Author response: Thank you reviewing our corrections to Reviewer 2's comments and for handling our manuscript fairly and efficiently.*

Residual reviewer comment:

It is my second time handling this paper. I was reviewer #1 the previous time.
I have checked the authors' responses to all comments raised by both reviewers and they are mostly satisfying. I recommend accepting the paper after authors perform a technical correction. They should clarify their explanation of the Gini impurity (lines 180-187). The way equation (1) is presented in the revision, the Gini impurity reads like a static number and it is not clear to readers how a feature changes it. I would recommend for authors to either add details to explain how the Gini impurity score is used in practice to compute feature importance, or to drop the equation.

*Author response: Thank you for this final amendment. We have dropped the equation as suggested. We have also shortened and tidied up the description of the feature importance computation to provide the required high-level information on this approach. Gini impurity is used widely in classification and references are provided for readers that wish to replicate our classification analysis.*